# Targeted Outreach by an Insurance Company Improved Dietary Habits and Urine Sodium/Potassium Ratios Among High-Risk Individuals with Lifestyle-Related Diseases

**DOI:** 10.3390/nu17132152

**Published:** 2025-06-27

**Authors:** Sunao Tanaka, Junji Fukui, Akira Otsu, Shintaro Yokoyama, Tsukasa Tanaka, Kaori Sawada, Shigeyuki Nakaji, Yoshinori Tamada, Koichi Murashita, Tatsuya Mikami

**Affiliations:** 1Innovation Centre for Health Promotion, Hirosaki University Graduate School of Medicine, Aomori 036-8562, Japan; tanakas@hirosaki-u.ac.jp (S.T.); shintaro-y@hirosaki-u.ac.jp (S.Y.); tnk_tks89@hirosaki-u.ac.jp (T.T.); nakaji@hirosaki-u.ac.jp (S.N.); 2Meiji Yasuda Life Insurance Company, Tokyo 100-0005, Japan; j-fukui@meijiyasuda.co.jp (J.F.); ak-otsu@meijiyasuda-group.com (A.O.); 3Department of Preemptive Medicine, Hirosaki University of Graduate School of Medicine, Aomori 036-8562, Japan; iwane@hirosaki-u.ac.jp; 4Research Centre for Health Medical Data Science, Hirosaki University Graduate School of Medicine, Aomori 036-8562, Japan; y.tamada@hirosaki-u.ac.jp; 5Research Institute of Health Innovation, Hirosaki University Graduate School of Medicine, Aomori 036-8562, Japan; murasita@hirosaki-u.ac.jp

**Keywords:** urine sodium/potassium ratio, health promotion, collaboration with university medical school and private insurance company, lifestyle habits, high-risk and treating customers

## Abstract

**Background/Objectives**: The urine sodium/potassium (Na/K) ratio can potentially be used to detect dietary habits that contribute to hypertension. In this prospective cohort interventional trial, we aimed to verify whether private insurance sales staff can help clients change their lifestyle habits based on their urinalysis results. **Methods:** Clients of the life insurance company (20–65 years old) who were considered to have “high risk” lifestyle factors, which was defined as having high values for two or more of the following indicators: body mass index, blood pressure, triglycerides, liver enzymes, and glucose metabolism, were included. The clients were randomly assigned to three groups: a face-to-face (FF) intervention by sales staff (n = 83), non-FF (Non-FF) intervention via a social networking service (n = 87), and no intervention (Control) (n = 58). Urinalysis and surveys about diet and exercise habits were conducted before and after a 3-month interventional period in all groups. Three interventions were performed for the FF and Non-FF groups, including dietary advice based on urinalysis results, education encouraging reduced salt intake and increased locomotor activity, and viewing an educational video. The Control group only received their urinalysis results by mail. **Results:** The participants’ mean age was 44.0 years old. Significant improvements in estimated potassium intake were observed in the Non-FF group, and significant reductions in urine Na/K ratios were noted in both the FF and Non-FF groups. Multiple logistic regression analysis indicated that watching the video was the most effective factor for decreasing the urine Na/K ratio (odds ratio = 1.869). The total points for dietary behavior, based on the questionnaire, significantly improved among the individuals who watched the video. **Conclusions:** This study demonstrates the potential for private health insurance companies to contribute to health promotion and introduces a novel strategy for improving lifestyle habits among individuals at high risk of lifestyle-related diseases.

## 1. Introduction

Poorly controlled hypertension is strongly associated with metabolic syndrome [1], and results in an increased risk of cardiovascular events. Hypertension affects approximately 45–49% of the US population [2], 43–48% of Japanese individuals [3], and more than 1 billion adults worldwide [2]. Several established nonpharmacological interventions for the prevention and treatment of hypertension have been reported [4], among which decreasing one’s sodium (Na) intake [5] and increasing one’s potassium (K) intake [6] are important factors for lowering blood pressure.

Recently, the urine Na/K ratio has been found to be more strongly associated with high blood pressure than the daily salt intake estimated from urinary Na [7]. Kogure M et al. [8] reported that the urinary Na/K ratio is a potential indicator for countering hypertension during health checkups in community settings. The urine Na/K ratio is a useful tool for self-monitoring hypertension-related dietary habits that does not require hospital visits. The Japanese Society of Hypertension recommends an average urine Na/K ratio of 4 as a feasible target value to achieve temporary goals in the Japanese general population [9]. Due to its simplicity and immediacy, the urine Na/K ratio has been adopted in health check-up programs that focus on quality of life in Japan [10].

Since 2008, Japanese health insurance holders aged 40–74 years have been legally encouraged to participate in the Specific Health Check-Ups and the Specific Health Guidance programs [11], which focuses on high visceral fat obesity. These programs utilize checkups that screen for symptoms of metabolic syndrome, and the Specific Health Guidance program is provided to individuals at risk for metabolic syndrome to improve their lifestyle habits. The number of people who are eligible for the Specific Health Check-Ups program in the fiscal year 2023 was approximately 52.1 million, and the number of people who actually underwent the checkups was approximately 31.23 million (59.9%). The number of people eligible for the Specific Health Guidance program was approximately 5.10 million, but the implementation rate was 28.6% (https://www.mhlw.go.jp/stf/seisakunitsuite/bunya/newpage_00063.html, accessed on 24 June 2025). The participants can access health information disseminated by the government mainly through leaflets and the internet, and they can also access health information provided by private broadcast media; of course, whether or not they access health information is left to their discretion.

Currently, most individuals can receive health information through social networking sites (SNSs) [12]. A key health promotion task is determining how and by whom correct health knowledge should be conveyed to the public. Although medical schools are generally considered reliable institutions for accessing current and academic health-related knowledge, many organizations lack the sufficient capacity to disseminate health knowledge to the public. Workplaces [13], schools [14], and local communities connected by common interests [15] are effective health promotion communities. In recent years, cooperation between public institutions and companies has been reported [16]; however, few reports exist on collaborations between university medical schools and life insurance companies for health promotion.

Recently, private life insurance services have become increasingly common, offering incentives such as lower insurance premiums to individuals who participate in health promotion activities [17,18]. The financial incentive approach is expected to change habitual health-related behaviors [19]. Traditionally, life insurance sales staff in Japan have visited customers individually to provide information about insurance products and handle renewal procedures. It is expected that offering lifestyle improvement advice when sharing diagnostic results related to customers’ health status and information on their lifestyle habits will further enhance that effect.

In this study, a private health insurance company collaborated with a university medical school to conduct a prospective intervention trial among insurance clients with certain high-risk lifestyle factors or who were undergoing treatment for lifestyle-related diseases. The intervention involved either a sales staff visit or social networking notifications that encouraged individuals to read distributed leaflets and watch video programs aimed at improving eating and exercise habits. It was hypothesized that face-to-face visits by sales staff would encourage customers to improve their lifestyle habits. The objective was to determine the most effective intervention to change the behavior in individuals with lifestyle diseases at the preliminary stage.

## 2. Materials and Methods

### 2.1. Study Design and Settings

This was a 3-month, prospective, randomized controlled intervention study conducted from August 2022 to December 2022. This study was approved by the Ethics Committee of Hirosaki University School of Medicine (approval numbers: 2023-043, 2025-002; approval date: 26 December 2022, 8 April 2025) and registered in the Japanese Registry of Clinical Trials (clinical research plan No: jRCT1020220037; authorization number: CRB2210001).

The life insurance company Company M ranked insurance clients using its own classification based on Specific Health Check-Up results and provided premium cashback accordingly. The algorithm for determining a client’s “Cash-back rank” is shown in Appendix A. In brief, cash-back ranks 2 or 3 were considered to indicate “high risk” for lifestyle-related disease. This was defined as having levels of two or more of the following in the upper normal range with one or more values above normal: body mass index (BMI), blood pressure, triglycerides, liver enzymes, and glucose metabolism. In this study, clients ranked as ‘Cash-back rank’ 2 or 3, or those who had already started treatment, were eligible for inclusion.

Each sales staff member of Company M recruited participants from among their own clients who lived in Aomori Prefecture in Japan. Participants included individuals who were aged 18–65 years, who were subscribed to voluntary health and life combo insurance targeting lifestyle-related diseases that was provided by Company M, who were ranked as ‘Cash-back rank as 2 or 3′, and who were able to cooperate with surveys involving sales staff visits, social networking via smartphones, urinalysis, and questionnaires. The exclusion criteria included being a family member of company sales staff or having a history of previous diseases such as cancers, heart disease, cerebrovascular disease, diabetes, hypertensive retinopathy, chronic kidney disease, chronic hepatitis, cirrhosis of the liver, or chronic pancreatitis. Participants and sales staff received gift cards for their participation in the trial.

After providing informed consent, participants were randomly assigned to one of three different groups: face-to-face (FF), non-face-to-face (Non-FF), and non-intervention (Control). A total of 260 participants were enrolled. After excluding those who withdrew consent or failed to submit urinalysis results or complete questionnaires, 83, 87, and 58 participants in the FF, Non-FF, and Control groups, respectively, were included in the analysis (Figure 1).

### 2.2. Intervention Strategies

At baseline, all patients underwent urinalysis and completed a questionnaire. Urinalysis results were mailed to participants by the contractors hired to perform the analysis. The pre-interventional questionnaire collected information on sex, age, occupation, height, weight, past and present history of medical treatment for lifestyle-related diseases, recent suggestions for lifestyle-related diseases in the Specific Health Check-Ups, and lifestyle questionnaires on dietary behaviors and locomotor activities (Appendix A). Table 1 shows the questionnaire on lifestyle habits that was used to confirm changes in participants’ awareness before and after the intervention. Participants were asked to answer multiple-choice questionnaires regarding their lifestyle habits, rating them as follows: 1, applicable; 2, somewhat applicable; 3, undecided; 4, not very applicable; and 5, not applicable.

For the intervention groups (FF and Non-FF), health information was provided three times over 3 months. The first intervention involved dietary advice based on the urinalysis results, such as the appropriate salt intake. The second intervention involved providing two types of health information through leaflets (Appendix A). The first leaflet addressed eating habits, including recommendations to limit carbohydrate and salt intake and increase dietary fiber intake. The second leaflet addressed locomotor activities, provided information on locomotive syndrome [20], and recommended increasing walking and daily activities by 10 min. The third intervention encouraged participants to watch an online video (Appendix A). In this video, a doctor in a white coat presented two cases: one where arterial fibrillation detected during a health checkup was left untreated, resulting in a stroke with residual disabilities, and another where hypertension from youth was left untreated, which led to heart failure. This video highlighted the importance of seeking prompt treatment if abnormalities are identified during health checkups. Company M produced all leaflets and videos under specialist supervision. For the FF group, leaflets and video encouragement were provided and explained by Company M sales staff. For the Non-FF group, materials were delivered via SNS. At the beginning of the interventional period, the participants were advised to install the application software provided by Company M on their smartphones to measure their daily number of steps. No health information, other than urinalysis results, was provided to the Control group.

All participants underwent urinalysis and completed a questionnaire survey 3 months after the trial initiation (Appendix A). The post-trial questionnaire included questions about height and weight, a lifestyle questionnaire on dietary behaviors and locomotor activities (Table 1), questions about whether participants had read the leaflets or watched the video, and questions about whether they were influenced by the urinalysis, leaflets, or video. An overview of the intervention trial is shown in Figure 2.

The main outcomes were a decrease in the urine Na/K ratio and changes in the estimated salt and K intake calculated by urinalysis. The secondary outcomes were changes in lifestyle questionnaire scores, BMI, and steps per day.

### 2.3. Data Collection and Processing

The spot urine collection samples were analyzed using Shio-Check plus (Healthcare Systems Co., Ltd., Tokyo, Japan). The estimated salt intake was calculated by converting the estimated Na excretion, derived using Tanaka’s formula [21], into the corresponding amount of salt. The estimated K intake was calculated by dividing the estimated K excretion, based on Tanaka’s formula [21], by the excretion rate and converting it into intake. The formulas were as follows: (1) PRCr (mg/d) = −2.04 × age + 14.89 × weight (kg) + 16.14 × height (cm) −2244.45; (2) estimated 24 h urinary sodium (mEq/d) = 21.98 × XNa (0.392); (3) estimated 24 h urinary pottasium (mEq/d) = 7.59 × XK (0.431), where PRCr = predicted value of 24 h urinary creatine, SUNa = Na concentration in the spot voiding urine, SUK = K concentration in the spot voiding urine, SUCr = creatinine concentration in the spot voiding urine, XNa (or XK) = SUNa (or SUK)/SUCr × PRCr. The urinary Na/K ratio was calculated by dividing the measured Na value by the measured K value. This method has been described as not suitable for estimating individual values of Na and K excretion, but has been considered useful for estimating population mean levels of 24 h Na and K excretion, and it is useful for comparing different populations [21].

Sales staff collated the urinalysis and questionnaire data and sent them to Company M, where they were anonymized and provided to the analysts. The height and weight obtained from the questionnaire responses were converted to BMI by dividing the weight (kg) by the square of the height (m) and were included in the analysis.

Participants answered multiple-choice questionnaires regarding lifestyle habits, and each response was assigned a point value. The total score for five questions on eating habits and five questions on locomotor activities, each on a 25-point scale, was compared before and after the intervention.

The average number of steps taken per day in the first and last weeks of the intervention was collected via application software and used in the analysis as the number of steps taken before and after the intervention.

### 2.4. Statistical Analyses

Differences among the three groups (FF, Non-FF, Control) were assessed using the Kruskal–Wallis test for the nominal variables and the chi-squared test for the continuous variables. The Wilcoxon signed-rank test was used to compare the estimated salt and K intakes and the urine Na/K ratio before and after the intervention within each group. Data from all participants were compared to evaluate changes in the urine Na/K ratio. Differences between two groups were analyzed using the Mann–Whitney U test for nominal variables and the chi-squared test for continuous variables. Multiple logistic regression analysis, using the likelihood ratio forward selection method, was performed to identify factors that influenced a decrease in the urine Na/K ratio. The dependent variable was whether the urine Na/K ratio decreased compared with baseline, and the covariates included the FF visits conducted by sales staff, whether participants were affected by the urinalysis, and whether they had read the leaflet or watched the video, based on responses to the post-intervention questionnaire. These covariates corresponded to types of interventions implemented in this trial. Multicollinearity was assessed before the independent variables were selected. In addition, two-group comparisons between participants with and without the extracted factors were conducted using the chi-squared test or the Mann–Whitney U test, and changes before and after the intervention were compared using the Wilcoxon signed-rank test.

All statistical analyses were performed using IBM SPSS Statistics software version 26.0 (IBM Corporation, Armonk, NY, USA). *p* values < 0.05 were considered statistically significant.

## 3. Results

### 3.1. Baseline Characteristics of the Three Groups

Table 2 presents the occupations of the participants. Many of the participants were desk workers, salespeople, service workers, and transport workers, and none were primary industry workers, such as farmers, forestry workers, or fishery workers.

The baseline characteristics of the three groups are summarized in Table 3. There were no significant differences in sex, age, Na and K intake, urine Na/K ratio, number of present and past treatments for lifestyle-related diseases, or number of previous suggestions received for hypertension. The mean (± standard deviation) BMI values were 25.2 ± 4.5, 24.3 ± 4.7, and 26.0 ± 4.0 kg/m^2^ in the FF, Non-FF, and Control groups, respectively, with significant differences being observed among these three groups. The number of participants with a urine Na/K ratio < 2.0, the optimal target value in Japan [9], was 16 (7.0%), and the number with a urine Na/K ratio < 4.0, a feasible target value in Japan [9], was 93 (40.7%). There were no significant differences in the total lifestyle questionnaire scores among the three groups. The numbers and percentages of participants who cooperated with step counting were 33 (39.7%), 46 (52.8%), and 26 (44.8%) in the FF, Non-FF, and Control groups, respectively; no significant differences were observed in the steps per day before the intervention.

### 3.2. Changes in Estimated Salt and Potassium Intake and the Urine Na/K Ratio

The changes in the estimated salt and K intake and in the urine Na/K ratio are shown in Figure 3. The K intake relatively significantly increased from 1743.0 ± 386.8 mg to 1935.8 ± 521.8 mg in the Non-FF group (*p* < 0.01). The urine Na/K ratios significantly decreased from 5.10 ± 2.93 to 4.35 ± 2.56 in the FF group (*p* < 0.05) and from 5.39 ± 3.08 to 4.65 ± 3.07 in the Non-FF group (*p* < 0.05). The estimated salt intake did not change significantly in any group. No significant differences were observed in the Control group.

### 3.3. Factors Associated with a Decrease in the Urine Na/K Ratio

To investigate the factors associated with decreases in the urine Na/K ratio, differences between the baseline and questionnaire responses were compared, and the participants were divided into decreased and non-decreased Na/K ratio groups (Table 4). There were no significant differences in terms of the participants’ interventional group, age, sex, BMI, past or present treatment of lifestyle-related diseases, recent suggestion of hypertension, or total lifestyle questionnaire scores. A total of 224 (98.2%) participants completed the post-interventional questionnaire. Of these, 160 (71.4%) reported reading the leaflets almost entirely or partially and 132 (58.9%) reported watching the video almost entirely or partially. A significant difference was observed in the number of individuals who watched the video between the two groups (*p* < 0.05). No significant differences were noted in the content that participants felt had affected them.

Multiple logistic regression analysis indicated that watching educational videos was significantly associated with a decrease in the urine Na/K ratio (Table 5). The odds ratio was 1.869, with a significant chi-squared test result (*p* < 0.05), and the variables were also significant (*p* < 0.05). The discriminant accuracy was 58.5%, which was not high.

### 3.4. Differences Between the Participants Who Watched or Did Not Watch the Educational Video

Table 6 presents changes among the participants who watched the educational video. A total of 132 participants reported watching the video. The estimated K intake significantly increased from 1786.4 ± 401.5 to 1920.3 ± 506.4 mg (*p* < 0.05), and the urine Na/K ratio significantly decreased from 5.28 ± 2.85 to 4.49 ± 2.83 mg (*p* < 0.01) among these participants. The total points for dietary behavior significantly increased from 13.3 ± 4.3 to 14.9 ± 2.7 (*p* < 0.05). The total points for locomotor activity significantly decreased from 14.6 ± 5.0 to 13.9 ± 4.6 (*p* < 0.05). The BMI and number of steps per day showed no significant changes.

Table 7 presents differences between the participants who watched and those who did not watch the educational video. There were significant differences in BMI at baseline (*p* < 0.05) or after the intervention period (*p* < 0.05). There were no significant differences in terms of the participants’ age, sex, past or present treatment of lifestyle-related diseases, recent suggestion of hypertension or other lifestyle diseases, or type of intervention.

## 4. Discussion

In this study, three groups were established: FF, Non-FF, and Control. It was hypothesized that FF interventions by life insurance company sales staff would improve the lifestyle habits of customers at high risk for lifestyle-related diseases or those undergoing initial treatment. The estimated K intake significantly increased in the Non-FF group, and the urine Na/K ratios significantly decreased in both the FF and Non-FF groups (Figure 3). Therefore, the interventions that were most effective in decreasing urine Na/K ratios were considered next. Univariate analysis was performed to compare the urine Na/K ratios between the decreased and the non-decreased groups, and watching the video was identified as a significant factor. The odds ratio for watching the video, adjusted for the presence or absence of FF involvement by sales staff and leaflet reading, was 1.869. Among the population who watched the video, the estimated K intake, urine Na/K ratio, and total dietary behavior scores significantly improved after the intervention (Table 6). These results indicate that watching an awareness-raising video is the most effective intervention for improving dietary habits. This report demonstrates that an intervention conducted by a private health insurance company under the supervision of a university medical school can contribute to the improvement of customers’ lifestyle habits.

At baseline, a significant difference in BMI was observed among the three groups (Table 3). The average BMI was highest in the Control group (26.0) and lowest in the Non-FF group (24.3), which suggested a potential inappropriate allocation at the initiation of the intervention trial. Therefore, the BMIs of the participants were retrospectively compared based on the results of this study. The baseline BMI of those who did not watch the educational video (25.8 ± 4.2) was significantly higher than that of those who watched the video (24.5 ± 4.6) (*p* < 0.05) (Table 7). These findings suggest that individuals without obesity were more aware of lifestyle improvements and may be more likely to benefit from interventions than individuals with obesity. A longitudinal investigation of children from childhood to adulthood reported significant positive associations among their urine Na/K ratios, blood pressure, and abdominal obesity, respectively [22]. Recently, the urine Na/K ratio has been shown to correlate more strongly with blood pressure than with salt intake [7] and has been reported as a useful indicator of diet quality in population-based studies [23]. The urinary Na/K ratio is thus considered a representative indicator of dietary habits, not only the estimated salt and K intakes, and individuals who are attentive to their dietary habits are more likely to not have obesity. Conversely, individuals with obesity who are not engaged in improving their eating habits are likely to be indifferent to lifestyle improvements; thus, targeting these groups represents an ongoing challenge in health promotion.

The participants who watched the educational video showed an increased estimated K intake and a significant decrease in their urine Na/K ratio. These results suggest that many individuals who watched the educational video increased their consumption of vegetables and/or fruits and improved their dietary habits. In contrast, the BMI, total locomotor activity points, and number of steps per day did not show significant changes. A previous systematic review revealed that dietary behavior was more likely to change than physical activity as a result of telephone interventions [24]. Eating habits may be easier to improve than exercise habits, particularly in the short term. Additionally, combining assessments such as the urine Na/K ratio, which is easily quantifiable, may help motivate individuals to improve their dietary habits.

The optimal target for the urine Na/K ratio is < 2.0 [9]; however, only 16 participants (7.0%) met this standard value at baseline. As Japanese individuals traditionally consume more Na, the feasible target value for this population was set at < 4.0 [9]. Across the entire population in this trial, the number of participants with a urine Na/K ratio < 4.0 increased from 93 (40.7%) at baseline to 118 (51.7%) after the intervention, although 72.7% of the participants reported that the urinalysis had an impact. Urine Na/K ratio analysis thus has the potential to raise awareness regarding the need for dietary improvement and to sensitively reflect these changes.

According to the 2023 “National Health and Nutrition Survey” report released by Japan’s Ministry of Health, Labour and Welfare (https://www.mhlw.go.jp/stf/seisakunitsuite/bunya/kenkou_iryou/kenkou/eiyou/r5-houkoku_00001.html (accessed on 24 June 2025)), the median salt intake (expressed as sodium equivalent) in Japan by age group (20–29, 30–39, 40–49, 50–59 years) was 8.9 g, 8.9 g, 9.3 g, and 9.2 g, respectively, while the K intake was 1914 mg, 1953 mg, 1952 mg, and 2102 mg, respectively. However, Healthcare Systems Co., Ltd. reported the average estimated salt intake in Japanese women (n = 8867) to be 8.70 ± 1.86 g/d, the estimated K intake to be 1871 ± 489 mg/d, and the urine Na/K ratio to be 3.84 ± 2.05 (mean ± standard deviation; S.D.) (n = 8867). The average values for the Tohoku region, including Aomori Prefecture, which was the area included in this study, were 8.94 ± 1.87 g, 1908 ± 416 mg, and 3.92 ± 2.03 (mean ± standard deviation), respectively (n = 384) (https://hc-sys.com/news/press-release/211119/). Although direct comparisons are not possible due to differences, the baseline characteristics of the participants in this study were estimated to reflect a slightly higher salt intake and lower K intake compared to the national average in Japan, and the mean urine Na/K ratio was higher than the national value by more than 1 (Table 3). The Tohoku region has been one of the areas in Japan with the highest salt intakes. The results of this study indicate that, despite interventions, the average estimated salt intake did not decrease. This fact suggests that it is challenging to change taste preferences acquired in infancy within a short period [25].

The total locomotor activity scores showed a significant decrease after the intervention in the video-watching group. This result may be attributable to the trial being conducted from autumn to winter in northern Japan. Physical activity levels and step counts tend to decrease during winter in Japan [26], particularly in northern regions [27]. However, the trial duration and intervention methods were insufficient to improve participants’ physical activity.

The questionnaire used in this study has not been scientifically validated. However, the questionnaire we used was in line with general guidelines for questionnaire design [28]. We used a question format and avoided yes/no questions for two-choice questions, labeled each option with words, used positive phrasing, used five options, and maintained visual spacing consistency. The results obtained from the questionnaire showed consistent trends that were in line with the urinalysis findings. However, further verification of its validity is necessary, including reliability and validity assessments as well as factor analysis.

This study has some limitations. The target population did not include primary industry workers, such as those in agriculture, forestry, or fishing, and was biased toward individuals who were approximately 40 years of age (18–65 years). The trial was also limited to clients of a single life insurance company in a prefecture in northern Japan. The participants’ BMI was self-reported and, therefore, may not have been accurate. In this study, all participants and sales staff received gift cards for trial participation, and it cannot be excluded that the financial incentive may have influenced the results. The questionnaire assessing lifestyle habits and the scoring method were developed independently and lacked external validation. Blood pressure was not investigated, as it was anticipated that no changes would occur within the 3-month intervention period. It is worth noting that the results of this study may not be applicable to individuals living in countries with different health checkup and insurance enrollment systems or to individuals who do not have health and/or life insurance. Future research should examine methods to ensure the long-term maintenance of improved dietary habits that lead to the normalization of blood pressure. Interventions by non-certified sales staff depend on the quality of individual skills, and interventions through leaflets and videos are likely influenced by the content provided.

## 5. Conclusions

An intervention by a private health insurance company could improve the urine Na/K ratios and dietary habits in clients who are at a high risk for, or undergoing treatment for, lifestyle-related diseases. Watching educational videos emerged as the most significant factor in achieving these improvements. This study proposes a novel approach to enhancing lifestyle habits in individuals at high risk for lifestyle-related diseases.

## Figures and Tables

**Figure 1 nutrients-17-02152-f001:**
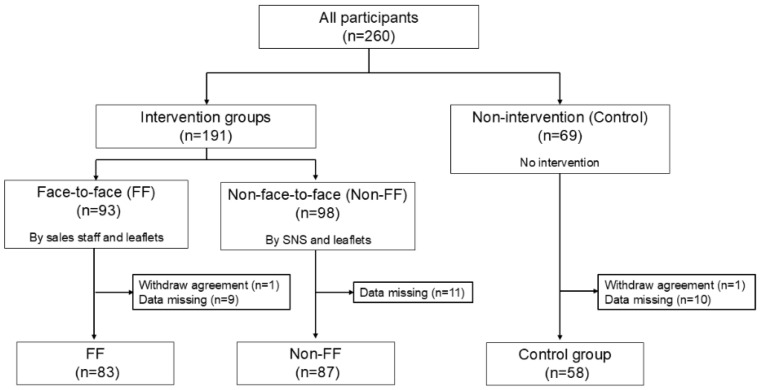
Flowchart of the classifications of participants.

**Figure 2 nutrients-17-02152-f002:**
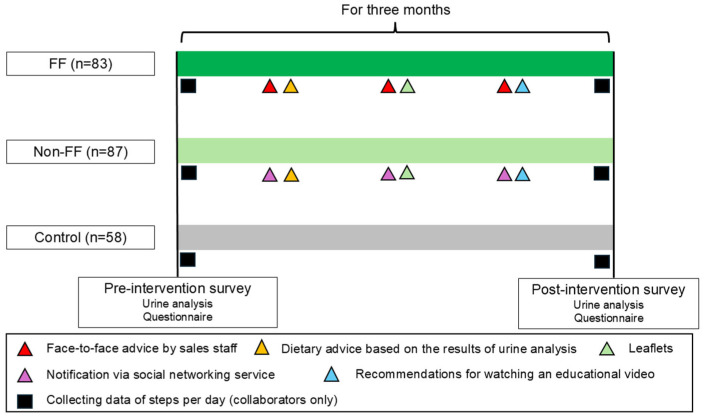
Schematic diagram of an interventional trial. FF: face-to-face intervention group, Non-FF: non-face-to-face intervention group, Control: control group (no-intervention group).

**Figure 3 nutrients-17-02152-f003:**
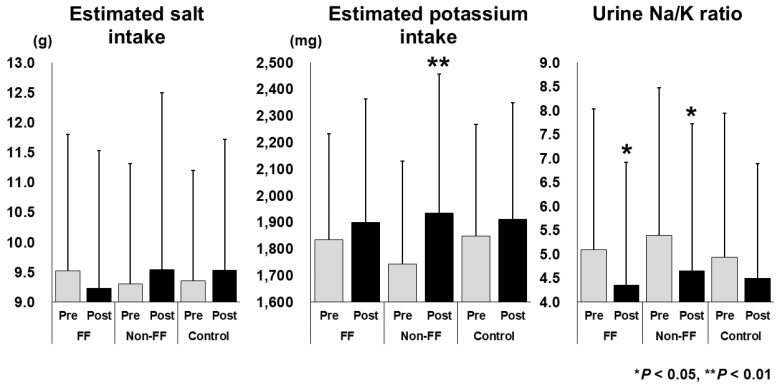
The results of the urinary analysis. Pre: pre-intervention, Post: post-intervention, FF: a face-to-face intervention group, Non-FF: non-face-to-face intervention group, Control: control group (no-intervention group). * *p* < 0.05, ** *p* < 0.01.

**Table 1 nutrients-17-02152-t001:** Lifestyle habit questionnaire items.

Dietary behaviors
I’m careful not to avoid excessive energy intake.
I’m careful not to eat too much salt.
I’m careful not to get too fat.
I try to consume an adequate intake of potassium, vitamins, minerals, and fiber.
I try to maintain an appropriate carbohydrate intake.
Locomotor activities
I perform regular gymnastics/stretching.
I’m aware of standing up, moving, standing tall, etc., regularly.
I regularly do aerobic exercise (walking, running, and aquabics, etc.).
I usually do strength training.
I try to incorporate a little exercise into my daily life, such as using the stairs or doing squats during breaks.

**Table 2 nutrients-17-02152-t002:** Occupations of participants.

	FF (n = 83)	Non-FF (n = 87)	Control (n = 58)
Professional/technical workers	18 (21.7%)	21 (24.1%)	14 (24.1%)
Manager	5 (6.0%)	1 (1.1%)	5 (8.6%)
Clerk	16 (19.3%)	19 (21.8%)	15 (25.9%)
Sales worker	3 (3.6%)	11 (12.6%)	3 (5.2%)
Service worker	10 (12.0%)	17 (19.5%)	8 (13.8%)
Security worker	2 (2.4%)	1 (1.1%)	3 (5.2%)
Construction and civil engineering	4 (4.8%)	2 (2.3%)	1 (1.7%)
Agriculture	0 (0.0%)	0 (0.0%)	0 (0.0%)
Forestry	0 (0.0%)	0 (0.0%)	0 (0.0%)
Fishing	0 (0.0%)	0 (0.0%)	0 (0.0%)
Transport and communications	7 (8.4%)	9 (10.3%)	3 (5.2%)
Production process and laborers	7 (8.4%)	0 (0.0%)	4 (6.9%)
Housewives	0 (0.0%)	1 (1.1%)	0 (0.0%)
None	2 (2.4%)	1 (1.1%)	0 (0.0%)
Others	9 (10.8%)	4 (4.6%)	2 (3.4%)

Data are presented as the number of individuals (the percentage).

**Table 3 nutrients-17-02152-t003:** Baseline characteristics of participants. Values are presented as means ± standard deviations.

	FF (n = 83)	Non-FF (n = 87)	Control (n = 58)	*p* Value
Male	54 (65.0%)	50 (57.4%)	41 (70.6%)	0.253
Age (years)	43.9 ± 10.0	43.4 ± 10.9	43.4 ± 10.1	0.983
BMI (kg/m^2^)	25.2 ± 4.5	24.3 ± 4.7	26.0 ± 4.0	0.041 *
Estimated salt intake (g/day)	9.52 ± 2.28	9.30 ± 2.00	9.35 ± 1.84	0.942
Estimated potassium intake (mg/day)	1835.5 ± 396.5	1743.0 ± 386.8	1849.2 ± 419.0	0.167
Urine Na/K ratio	5.10 ± 2.93	5.39 ± 3.08	4.94 ± 3.00	0.658
<2.0	8 (9.6%)	3 (3.4%)	5 (8.6%)	
≧2.0, <4.0	27 (32.5%)	30 (34.5%)	20 (34.5%)	
≧ 4.0	48 (57.8%)	54 (62.1%)	33 (56.9%)	
Present treatment of lifestyle disease	20 (24.1%)	22 (25.3%)	20 (34.5%)	0.347
Past treatment of lifestyle disease	21 (25.3%)	22 (25.3%)	17 (29.3%)	0.835
Previous suggestions of hypertension	16 (19.3%)	22 (25.3%)	15 (25.9%)	0.560
Total scores in the lifestyle questionnaire				
Dietary behavior	13.1 ± 4.6	13.6 ± 4.4	13.7 ± 4.5	0.624
Locomotor activity	15.0 ± 5.0	14.7 ± 4.9	15.0 ± 4.8	0.328
Number of participants who cooperated with step counting	33 (39.7%)	46 (52.8%)	26 (44.8%)	
Steps per day	3825.8 ± 2446.8	4876.9 ± 3080.8	4397.0 ± 2970.2	0.328

Data are presented as the number of individuals (the percentage of individuals in the group) or means ± standard deviation. BMI, body mass index. * *p* < 0.05.

**Table 4 nutrients-17-02152-t004:** Difference between groups with decreased and non-decreased urinary sodium/potassium (Na/K) ratios.

		Urine Na/K Ratio		
		Decreased (n =128)	Not Decreased (n = 100)	*p* Value
Interventional groups				
	FF	51 (39.8%)	32 (32.0%)	
	Non-FF	49 (38.3%)	38 (38.0%)	0.300
	Control	28 (21.9%)	30 (30.0%)	
Age		43.0 ± 10.6	44.3 ± 9.9	0.324
Sex				
	Male	82 (64.1%)	63 (63.0%)	0.869
	Female	46 (35.9%)	37 (37.0%)	
BMI (kg/m^2^)		24.8 ± 4.4	25.3 ± 4.6	0.478
Past treatment of lifestyle disease		29 (22.7%)	31 (31.0%)	0.156
Present treatment of lifestyle disease		33 (25.8%)	29 (29.0%)	0.588
Recent suggestion of hypertension		30 (23.4%)	23 (23.0%)	0.938
Total scores in lifestyle questionnaire (baseline)				
	Dietary behaviors	13.2 ± 4.4	13.8 ± 4.6	0.435
	Locomotor activities	15.1 ± 5.0	14.6 ± 4.9	0.382
The number of participants who answered the post-interventional questionnaire		224	224	
Whether the leaflets had been read				
	Almost all/partially	92 (41.1%)	68 (30.4%)	0.458
	None	33 (14.7%)	31 (13.8%)	
Whether the video had been watched				
	Almost all/partially	82 (36.6%)	50 (22.3%)	0.029 *
	None	43 (19.2%)	49 (21.9%)	
Affected by				
	Urinalysis	96 (42.9%)	67 (29.9%)	0.134
	Leaflets	63 (28.1%)	46 (20.5%)	1.000
	Video	58 (25.9%)	37 (16.5%)	0.842

Data are presented as the number of individuals (the percentage of individuals in the total number) or means ± standard deviation. BMI, body mass index. * *p* < 0.05.

**Table 5 nutrients-17-02152-t005:** Results of the multiple logistic regression analysis.

	Constant	Partial Regression Coefficient	Significant Probability	Odds Ratio	95% Confidence Interval
Watched the educational video	–0.131	0.625	0.023 *	1.869	1.089–3.206

* *p* < 0.05.

**Table 6 nutrients-17-02152-t006:** Changes among participants who watched the educational video.

	Pre-Intervention	Post-Intervention	*p* Value
Estimated salt intake (g/d)	9.44 ± 2.00	9.36 ± 2.61	0.482
Estimated potassium intake (mg/d)	1786.4 ± 401.5	1920.3 ± 506.4	0.007 *
Urine Na/K ratio	5.28 ± 2.85	4.49 ± 2.83	0.001 **
BMI (kg/m^2^)	24.5 ± 4.6	24.6 ± 4.6	0.476
Total points of dietary behavior	13.3 ± 4.3	14.9 ± 3.7	<0.001 **
Total points of locomotor activity	14.6 ± 5.0	13.9 ± 4.6	0.045 *
Numbers of individuals who answered the question on the number of steps	63 (47.4%)	59 (44.6%)	
Steps per day	4542.9 ± 2940.6	4483.7 ± 3124.3	0.590

Data are presented as the number of individuals (the percentage of individuals in the total number) or means ± standard deviation. BMI, body mass index. * *p* < 0.05, ** *p* < 0.01.

**Table 7 nutrients-17-02152-t007:** Differences between participants who watched and did not watch the educational video.

	Watched (n = 132)	Did Not Watch (n = 92)	*p* Value
Age	44.1 ± 10.7	42.8 ± 9.7	0.305
Sex			
Male	82 (62.1%)	61 (66.3%)	0.573
Female	50 (37.9%)	31 (33.7%)	
BMI (baseline) (kg/m^2^)	24.5 ± 4.6	25.8 ± 4.2	0.018 *
BMI (after the intervention period) (kg/m^2^)	24.6 ± 4.6	25.8 ± 4.3	0.021 *
Past treatment of lifestyle disease	31 (23.5%)	25 (27.2%)	0.530
Recent suggestion of hypertension	28 (21.2%)	22 (23.9%)	0.369
Recent suggestion of other lifestyle diseases	63 (47.7%)	55 (59.8%)	0.530
The type of intervention			0.075
Face-to-face	62 (47.0%)	20 (21.7%)	
Non-face-to-face	70 (53.0%)	14 (15.2%)	0.251
Recent suggestion of hypertension	28 (21.2%)	22 (23.9%)	

Data are presented as the number of individuals (the percentage of individuals in the group) or means ± standard deviation. BMI, body mass index. * *p* < 0.05.

## Data Availability

Hirosaki University School of Medicine and Meiji Yasuda Life Insurance Company retained the raw datasets presented in this study but made them unavailable due to obligations related to protecting participant data. Requests to access the datasets were directed to tanakas@hirosaki-u.ac.jp.

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
