# Peer review of "Targeted Outreach by an Insurance Company Improved Dietary Habits and Urine Sodium/Potassium Ratios Among High-Risk Individuals with Lifestyle-Related Diseases"

_nutrients, 2025, doi:10.3390/nu17132152_

Round 1

Reviewer 1 Report

Comments and Suggestions for Authors

Dear Authors,

This is an interesting article on a topic of great importance to public health. The following comments are intended to strengthen the manuscript and make it more readable for a wider audience.

Abstract: In the title, the authors indicate that the study participants are "high risk". Therefore, the abstract should include further information that more accurately reflects the group. Could you please clarify on what basis it was assessed that the participants were at high risk of diet-related diseases? Could you please provide the ages of the participants?

Introduction: The objective of the work is not entirely clear to me: was the aim to assess factors influencing the pro-health attitudes of participants or to determine the most effective intervention? The final sentence of this section deals with the results. I believe that this element is not aligned with the introduction, and I would therefore suggest removing it. I am curious to know whether the participants had access to additional resources pertaining to healthy diets and lifestyles. In many countries, such campaigns are carried out by national media and reach a wide audience, e.g. through television or radio. The authors describe such activities in the section on methodology, but it seems to me that they should rather be placed in the introduction to give a better picture for a reader.

The authors could explain in more detail how the intake of these nutrients is assessed based on the Na/K ratio.

Materials and Methods: Why were individuals between the ages of 18 and 65 included in the study? It seems that young people are not a risk group for hypertension. Such a large age range causes a large diversity of the group.

The structure of the questionnaire regarding questions about life style habits is not clear (table 1). What does "awareness" mean - was knowledge/awareness checked objectively or were participants asked directly about their subjective feelings? It is worth clarifying.

I also wonder about the nutritional education provided. If the participants' dietary habits were not assessed, then on what basis was it assumed that, for example, they should be educated about limiting carbohydrates (line 128). This is not related to the assessed ratio of potassium to sodium in urine (which explains the education about salt or potassium intake).Was the education of participants individualised, e.g. if BMI indicated obesity, was this taken into account in the education?

Results: I believe that precisely listing the participants' professions does not add much to the manuscript. It might be better to provide your education, as this may influence the reception and understanding of educational materials.

Could the authors clarify the sodium and potassium intake values ​​given in Table 3 in mg. Is this the daily intake? The value of approximately 9 mg sodium per day is virtually impossible to achieve with a normal diet. It is also lower than the recommended intake. And a potassium intake of 3,500 mg/day is considered adequate for the adult population by EFSA/WHO, and the intake provided in table is less then 2000mg? Could the Authors comment on this. What is the typical intake in local population?

 In the control group, all participants had prior recommendations for hypertension, whereas in the other groups, this applied to less than half of the group. Do the authors think this could have influenced the study results?

Discussion: This part is clear and logical. However, I have reservations about the presented values ​​of sodium and potassium intake, which is why it is difficult for me to fully relate to the data discussed by the authors.

I am also wondering about the effectiveness of the intervention: potassium intake among participants who watched the educational video increased, but sodium intake did not decrease. The authors explain this by an increase in the consumption of vegetables and fruits as good sources of potassium. How, however, can the lack of change in sodium intake be explained? Is sodium intake or the sodium to potassium urinary ratio more important as a cardiovascular risk factor? Could the Authors comment on this.

Author Response

Replies to Reviewer’s comments (Reviewer 1)

 We are grateful to Reviewer #1 for the critical comments and useful suggestions that have helped us to improve our paper. As indicated in the responses that follow, we have taken into account the revised version of our paper.

Comment #1.

Abstract: In the title, the authors indicate that the study participants are "high risk". Therefore, the abstract should include further information that more accurately reflects the group. Could you please clarify on what basis it was assessed that the participants were at high risk of diet-related diseases? Could you please provide the ages of the participants?

Response to comment #1.

 Thank you for your insightful comment. The definition of the “high risk” group for lifestyle-related diseases in this study was very complex and was defined by the life insurance company’s own algorithm. We added the algorithm for determining the risk of lifestyle disease (‘Cash-back rank’) in Supplemental Data 1. In brief, “high risk” for lifestyle-related disease was defined as having two or more of the following values above the upper normal range and one or more values above normal: body mass index (BMI), blood pressure, triglycerides, liver enzymes, and glucose metabolism. We have added this information to the materials and methods section, along with the age of the participants, in the abstract.

Abstract:

Methods: Clients of the life insurance company (20–65 years old) who were in considered to have “high risk” lifestyle factors, defined as having high values for two or more of the following indicators: body mass index, blood pressure, triglycerides, liver enzymes, and glucose metabolism were included. Clients were randomly assigned to three groups…

Results: The participants’ mean age was 44.0 years old. Significant improvements in estimated potassium intake were observed…

  1. Materials and Method

The life insurance Company M ranked insurance clients using its own classification based on Specific Health Check-up results and provided premium cashback accordingly. The algorithm for determining a client’s “Cash-back rank” is shown in Supplemental Data 1. In brief, Cash-back ranks 2 or 3 were considered to indicate “high risk” for lifestyle-related disease. This was defined as having levels of two or more of the following in the upper normal range with one or more values above normal: body mass index (BMI), blood pressure, triglycerides, liver enzymes, and glucose metabolism In this study, customers ranked as ‘Cash-back rank’ as 2 or 3, or those who had already started treatment, were eligible for inclusion.

Comment #2.

Introduction: The objective of the work is not entirely clear to me: was the aim to assess factors influencing the pro-health attitudes of participants or to determine the most effective intervention? The final sentence of this section deals with the results. I believe that this element is not aligned with the introduction, and I would therefore suggest removing it. I am curious to know whether the participants had access to additional resources pertaining to healthy diets and lifestyles. In many countries, such campaigns are carried out by national media and reach a wide audience, e.g. through television or radio. The authors describe such activities in the section on methodology, but it seems to me that they should rather be placed in the introduction to give a better picture for a reader. The authors should explain in more detail how the intake of these nutrients is assessed based on the Na/K ratio.

Response to comment #2.

Thank you for your insightful comment. We acknowledge that the description of the study's purpose in the introduction was unclear. The objective was to determine the most effective intervention to change behavior in individuals with lifestyle diseases at the preliminary stage. As you mentioned, the final sentence of this section deals with the results and has been removed; the sentence has been revised to be clearer.

In Japan, citizens can access health information disseminated by the government mainly through leaflets and the internet. Health information provided by broadcast media is primarily through private broadcast media, and is not disseminated by the government. Although the Specific Health Check-up and Health Guidance program is established based on Japanese law, these targets are not enforceable. Therefore, whether or not citizens access health information is left to their discretion. We have added one paragraph in the Introduction section and explained this.

We described more about the explanation of how the estimated intake of salt and potassium was calculated from the spot urine collection sample at ‘2. Materials and Methods, 2.3. Data collection and processing.’

  1. Introduction

…It was hypothesized that face-to-face visits by sales staff would encourage customers to improve their lifestyle habits. The objective was to determine the most effective intervention to change behavior in individuals with lifestyle diseases at the preliminary stage.

Introductio

Since 2008, Japanese health insurance holders aged 40–74 years have been legally encouraged to participate in the Specific Health Check-ups and the Specific Health Guidance program[11], focusing on high visceral fat obesity. This program utilized checkups that screen for symptoms of metabolic syndrome, and the Specific Health Guidance program is provided to individuals at risk for metabolic syndrome to improve lifestyle habits. The number of people eligible for the Specific Health Check-ups in fiscal year 2023 was approximately 52.1 million, and the number of people who actually underwent the checkups was approximately 31.23 million (59.9%). The number of people eligible for specific health guidance was approximately 5.10 million, but the implementation rate was 28.6% (https://www.mhlw.go.jp/stf/seisakunitsuite/bunya/newpage_00063.html). The pparticipants can access health information disseminated by the government mainly through leaflets and the internet, and health information provided by private broadcast media; of course, whether or not they access health information is left to their discretion.

2.3. Data collection and processing

The spot urine collection sample was analyzed using Shio-Check plus (Healthcare Systems Co., Ltd., Japan). The estimated salt intake was calculated by converting the estimated Na excretion, derived using Tanaka's formula [21], into the corresponding amount of salt. The estimated K intake was calculated by dividing the estimated K excretion, based on Tanaka formula [21], by the excretion rate and converting it into intake. Formulas were as follows: (1) PRCr (mg/d) = -2.04 x age + 14.89 x weight (kg) + 16.14 x height (cm) -2244.45; (2) estimated 24-h urinary sodium (mEq/d) = 21.98 x XNa (0.392); (3) estimated 24-h urinary pottasium (mEq/d) = 7.59 x XK (0.431); where PRCr = predicted value of 24-h urinary creatine, SUNa = Na concentration in the spot voiding urine, SUK = K concentration in the spot voiding urine, SUCr = creatinine concentration in the spot voiding urine, XNa (or XK) = SUNa (or SUK)/SUCr × PRCr. The urinary Na/K ratio was calculated by dividing the measured Na value by the measured K value. This method has been described as not suitable for estimating individual values of Na and K excretion, but has been considered useful for estimating population mean levels of 24-h Na and K excretion, and it is available for comparing different populations[21].

Comment #3.

Materials and Methods: Why were individuals between the ages of 18 and 65 included in the study? It seems that young people are not a risk group for hypertension. Such a large age range causes a large diversity of the group. The structure of the questionnaire regarding questions about life style habits is not clear (table 1). What does "awareness" mean - was knowledge/awareness checked objectively or were participants asked directly about their subjective feelings? It is worth clarifying.

Response to comment #3.

Indeed, the prevalence of hypertension increases with age, but this does not mean that there are no young people with hypertension. The prevalence of hypertension has been reported to be 22.4% among adults aged 18–39 and increased to 54.5% among those aged 40–59, and 74.5% among those aged 60 and over in the United States (Ostchega, Y et al., Hypertension Prevalence Among Adults Aged 18 and Over: United States, 2017-2018, NCHS Data Brief 364, p1-8, 2020). Since there is no fixed age at which hypertension develops, we believe that examining the distribution of high Na/K ratios, which is a risk factor for hypertension, does not necessarily diversify the group.

 We wanted to measure changes in awareness of participants after the intervention. Table 1 was confusing because the items were not represented as a questionnaire, so we represented them as a questionnaire, and changed Table 1.

And we have added a sentence on this in the ‘Intervention strategies’ section. The questionnaire we used has not been scientifically validated. This is a limitation of the study, which we have addressed in the discussion section.

‘2.2 Intervention strategies’, paragraph 1

…The pre-interventional questionnaire collected information on sex, age, occupation, height, weight, past and present history of medical treatment for lifestyle-related diseases, recent suggestions for lifestyle-related diseases in the Specific Health Check-ups, and lifestyle questionnaires on dietary behaviors and locomotor activities (Supplemental Data 2). Table 1 shows the questionnaire on lifestyle habits used to confirm changes in participants’ awareness before and after the intervention. Participants were asked to answer multiple-choice questionnaires regarding lifestyle habits, rating them as: 1, applicable; 2, somewhat applicable; 3, undecided; 4, not very applicable; and 5, not applicable.

Table 1. Lifestyle habit questionnaire items.

Dietary behaviors

I’m careful not to avoid excessive energy intake.

I’m careful not to eat too much salt.

I’m careful not to get too fat.

I try to consume an adequate intake of potassium, vitamins, minerals, and fiber.

I try to maintain an appropriate carbohydrate intake.

Locomotor activities

I perform regular gymnastics/stretching.

I’m aware of standing up, moving, standing tall, etc., regularly.

I regularly do aerobic exercise (walking, running, and aquabics, etc.).

I usually do strength training.

I try to incorporate a little exercise into my daily life, such as using the stairs or doing squats during breaks.

Discussion

The questionnaire used in this study has not been scientifically validated. However, the questionnaire we used was in line with general guidelines for questionnaire design[28]. It used a question format and avoided yes / no questions for two-choice questions, labeled each option with words, used positive phrasing, used five options, and maintained visual spacing consistency. The results obtained from the questionnaire showed consistent trends that were in line with the urinalysis findings. However, further verification of its validity is necessary, including reliability and validity assessments as well as factor analysis.

Comment #4.

I also wonder about the nutritional education provided. If the participants' dietary habits were not assessed, then on what basis was it assumed that, for example, they should be educated about limiting carbohydrates (line 128). This is not related to the assessed ratio of potassium to sodium in urine (which explains the education about salt or potassium intake). Was the education of participants individualised, e.g. if BMI indicated obesity, was this taken into account in the education?

Response to comment #4.

Education on dietary habits was not individualized according to each person’s condition. This study initially expected that dietary habit improvement advice, such as avoiding excessive carbohydrate intake, would improve BMI even in people with high BMI, but no such results were obtained. Since the expected results were not demonstrated, this paper does not focus much on that point. As we mentioned in the Discussion, we believe that this is related to the use of the urine Na/K ratio as an objective test indicator, which reflects early changes in salt restriction.

Comment #5.

Results: I believe that precisely listing the participants' professions does not add much to the manuscript. It might be better to provide your education, as this may influence the reception and understanding of educational materials. Could the authors clarify the sodium and potassium intake values ​​given in Table 3 in mg. Is this the daily intake? The value of approximately 9 mg sodium per day is virtually impossible to achieve with a normal diet. It is also lower than the recommended intake. And a potassium intake of 3,500 mg/day is considered adequate for the adult population by EFSA/WHO, and the intake provided in table is less then 2000mg? Could the Authors comment on this. What is the typical intake in local population?

Response to comment #5.

Thank you very much for your keen insight. In this study, we surveyed participants’ occupations but did not survey their educational backgrounds. As you have pointed out, information on educational background may have been necessary to assess the level of understanding of lifestyle education. However, we also considered that the current occupation might be related to the ease of access to information, such as using social networking services or watching educational videos via the internet. Additionally, the survey results did not include individuals engaged in primary industries, which we chose to note. Therefore, we included information on participants’ occupations.

  There was a mistake in the unit of measurement for estimated salt intake. The correct unit was gram per day (g/d), not milligram (mg). We fixed the unit in Table 3, and Figure 3. Further, we noticed a lack of unit information in some figures, and have added unit labels in Table 4, Table 6, and Table 7. Further, we confused the sodium intake, we had been using the terms “sodium intake” and “salt intake” interchangeably, so we have corrected all instances of these terms throughout the manuscript.

Discussion

According to the 2023 “National Health and Nutrition Survey” report released by Japan's Ministry of Health, Labour and Welfare (https://www.mhlw.go.jp/stf/seisakunitsuite/bunya/kenkou_iryou/kenkou/eiyou/r5-houkoku_00001.html), the median salt intake (expressed as sodium equivalent) by age group (20–29, 30–39, 40–49, 50–59 years) was 8.9 g, 8.9 g, 9.3 g, and 9.2 g, respectively, while K intake was 1,914 mg, 1,953 mg, 1,952 mg, and 2,102 mg, respectively. However, Healthcare Systems Co., Ltd. reported the average estimated salt intake in Japanese women (n = 8,867) to be 8.70 ± 1.86 g/d, the estimated K intake to be 1,871 ± 489 mg/d, and the urine Na/K ratio to be 3.84 ± 2.05 (mean ± standard deviation; S.D.), (n = 8,867). The average values for the Tohoku region, including Aomori Prefecture, which was the area included in this study, were 8.94 ± 1.87 g, 1,908 ± 416 mg, and 3.92 ± 2.03 (mean ± standard deviation), respectively (n = 384) (https://hc-sys.com/news/press-release/211119/). Although direct comparisons are not possible due to differences, the baseline characteristics of the participants in this study were estimated to have slightly higher salt intake and lower K intake compared to the national average in Japan, and the mean urine Na/K ratio was higher than the national value by more than 1 (Table 3). The Tohoku region has been one of the areas in Japan with the highest salt intake. The results of this study indicate that, despite interventions, the average value of estimated salt intake did not decrease. This fact suggests that it is challenging to change taste preferences acquired in infancy within a short period[25].

Comment #6.

 In the control group, all participants had prior recommendations for hypertension, whereas in the other groups, this applied to less than half of the group. Do the authors think this could have influenced the study results?

Response to comment #6.

In Table 3, the number of individuals who had previously been suggested to have had hypertension in the Control group was 15, not 58. We apologize for this error made during the copying process.

Comment #7.

Discussion: This part is clear and logical. However, I have reservations about the presented values ​​of sodium and potassium intake, which is why it is difficult for me to fully relate to the data discussed by the authors. I am also wondering about the effectiveness of the intervention: potassium intake among participants who watched the educational video increased, but sodium intake did not decrease. The authors explain this by an increase in the consumption of vegetables and fruits as good sources of potassium. How, however, can the lack of change in sodium intake be explained? Is sodium intake or the sodium to potassium urinary ratio more important as a cardiovascular risk factor? Could the Authors comment on this.

Response to comment #7.

As we mentioned in our response to comment #5, there was an error in the units of measurement for estimated salt intake, and this has been corrected. We compared the estimated potassium intake with data from Japanese databases and data provided by the company that conducted the test, and we have added our considerations in the ‘Discussion’ section. The Tohoku region, including Aomori Prefecture where we conducted our survey, has been one of the areas in Japan with the highest salt intake. The results of this study indicate that, despite intervention, the average value of estimated salt intake did not decrease. This fact suggests that it is challenging to change taste preferences acquired in infancy within a short period. However, as mentioned in the introduction, recent studies have reported that the urine Na/K ratio is more strongly associated with high blood pressure than daily salt intake estimated from urinary Na. This suggests that improving the urine Na/K ratio may reduce the risk of hypertension in the future, which is the new insight of this paper. We would greatly appreciate it if you could take a moment to review the paper again, considering this point.

Reviewer 2 Report

Comments and Suggestions for Authors

This is a well-written, timely, and original article that addresses a novel approach to health promotion by leveraging private insurance infrastructure for dietary behavior change. The randomized controlled design and the collaborative framework between a university and a private insurance company are commendable. The study presents interesting data on sodium/potassium intake, behavioral change, and intervention efficacy.

Comments:

Emphasize more strongly in the introduction why this model (insurance-led outreach) fills a gap in existing health promotion approaches.

Please add the leaflets, the educational videos and the questionnaires as supplementary materials.

In Table 3, for the line “Previous suggestions of hypertension”, it says 58 for Controls, which means 100% of the Controls. Is that correct?

Please add percentages in the tables (out of n), it would be easier to see.

Were these questionnaires validated? Because the use of self-developed, non-validated lifestyle questionnaires reduces the reliability of behavior change measurement. Acknowledge this limitation more clearly in the discussion, and consider briefly describing the process of questionnaire development.

The population is limited to a specific age group (18–65), region (Aomori), and insurance company. Participants also received gift cards, which could bias engagement. Add a few lines about the limits of external validity, especially when generalizing to other countries or uninsured populations.

Author Response

Replies to Reviewer’s comments (Reviewer 2)

 We are grateful to Reviewer #2 for the critical comments and useful suggestions that have helped us to improve our paper. As indicated in the responses that follow, we have taken into account the revised version of our paper.

Comment #1.

Emphasize more strongly in the introduction why this model (insurance-led outreach) fills a gap in existing health promotion approaches.

Response to comment #1.

Thank you very much for your constructive comments on improving the quality of this paper. Based on your comment, we have added a paragraph to the introduction.

Introduction

Recently, private life insurance services have become increasingly common, offering incentives such as lower insurance premiums to individuals who participate in health promotion activities[17,18]. The financial incentive approach is expected to change habitual health-related behaviors[19]. Traditionally, life insurance sales staff in Japan have visited customers individually to provide information about insurance products and handle renewal procedures. It is expected that offering lifestyle improvement advice when sharing diagnostic results related to health status and information on customers’ lifestyle habits will further enhance that effect.

Comment #2.

Please add the leaflets, the educational videos and the questionnaires as supplementary materials.

Response to comment #2.

Thank you so much for your valuable feedback. We have newly added the questionnaires (Supplemental Data 2 and 5), leaflets (Supplemental Data 3), and a link to the educational video (Supplemental Data 4) to the supplementary materials.

Comment #3.

In Table 3, for the line “Previous suggestions of hypertension”, it says 58 for Controls, which means 100% of the Controls. Is that correct? Please add percentages in the tables (out of n), it would be easier to see. Were these questionnaires validated? Because the use of self-developed, non-validated lifestyle questionnaires reduces the reliability of behavior change measurement. Acknowledge this limitation more clearly in the discussion, and consider briefly describing the process of questionnaire development.

Response to comment #3.

Thank you very much for your keen insight. In Table 3, the number of individuals who had previously been suggested to have hypertension in the Control group was 15, not 58. This was simply an error made during the writing process. In response to your comment, we added percentages in Tables 2, 3, 4, and 7. As you pointed out, these questionnaires have been used conventionally by life insurance companies and have not been scientifically validated. We have now discussed this limitation and have described the process of questionnaire development in the discussion.

Discussion

The questionnaire used in this study has not been scientifically validated. However, the questionnaire we used was in line with general guidelines for questionnaire design[28]. It used a question format and avoided yes / no questions for two-choice questions, labeled each option with words, used positive phrasing, used five options, and maintained visual spacing consistency. The results obtained from the questionnaire showed consistent trends that were in line with the urinalysis findings. However, further verification of its validity is necessary, including reliability and validity assessments as well as factor analysis.

Comment #4.

The population is limited to a specific age group (18–65), region (Aomori), and insurance company. Participants also received gift cards, which could bias engagement. Add a few lines about the limits of external validity, especially when generalizing to other countries or uninsured populations.

Response for comment #4.

Thank you for these comments; we fully agree. We had already outlined the limitations regarding the targeted population (specific age group, region, and clients of a single insurance company) in the last paragraph of the discussion. Furthermore, we have now added new text that reflect your opinions.

Discussion

This study has some limitations. The target population did not include primary industry workers, such as those in agriculture, forestry, or fishing, and was biased toward individuals approximately 40 years of age (18–65 years). The trial was also limited to clients of a single life insurance company in a prefecture in northern Japan. BMI was self-reported by participants and therefore may not have been accurate. In this study, all participants and sales staff received gift cards for trial participation, and it cannot be excluded that the financial incentive may have influenced the results. The questionnaire assessing lifestyle habits and the scoring method were developed independently and lacked external validation. Blood pressure was not investigated, as it was anticipated that no changes would occur within the 3-month intervention period. It is worth noting that the results of this study may not be applicable to individuals living in countries with different health checkup and insurance enrollment systems or to individuals who do not have health and/or life insurance. Future research should examine methods to ensure the long-term maintenance of improved dietary habits, leading to the normalization of blood pressure. Interventions by non-certified sales staff depend on the quality of individual skills, and interventions through leaflets and videos are likely influenced by the content provided.

Round 2

Reviewer 1 Report

Comments and Suggestions for Authors

Thank to the Authors for addressing my comments

Reviewer 2 Report

Comments and Suggestions for Authors

The authors have responded well to my recommendations and have made all the necessary changes.